# Impact of the Clinical Pharmacist in Rheumatology Practice: A Systematic Review

**DOI:** 10.3390/healthcare12151463

**Published:** 2024-07-23

**Authors:** Eric Barat, Annaelle Soubieux, Pauline Brevet, Baptiste Gerard, Olivier Vittecoq, Thierry Lequerre, Catherine Chenailler, Rémi Varin, Claire Lattard

**Affiliations:** 1Department of Pharmacy, Rouen University Hospital, F-76000 Rouen, France; 2Department of Public Health, Normandie University, UNICAEN, Inserm U1086, F-14000 Caen, France; 3Department of Rheumatology & CIC-CRB 1404, CHU Rouen, University Rouen Normandie, UNIROUEN, F-76000 Rouen, France; 4Department of Rheumatology, Rouen University Hospital, F-76000 Rouen, France; 5Pharmacie des Alpes, F-38300 Domarin, France

**Keywords:** rheumatology, pharmaceutical care, care pathway

## Abstract

This is a systematic literature review on the impact of pharmacists in rheumatology, conducted using the PubMed^®^, CINAHL^®^, Cochrane Library^®^, and Web of science^®^ databases and using the PRISMA 2020 checklist. This review was conducted from 2000 to June 2024. A quality analysis was performed. The selection of articles, as well as all analyses, including quality analyses, were conducted by a pair of pharmacists with experience in rheumatology, and included 24 articles. This study highlights the growth of clinical pharmacy activities in rheumatology and the positive influence of clinical pharmacists on patient care. The implementation of such initiatives has the potential to improve medication adherence, reduce medication-related risks, and optimize associated healthcare costs. All these pharmaceutical interventions aim to make the patient care journey smoother and safer. Additionally, the diversity of available pharmaceutical services caters to the varied needs of rheumatology. Furthermore, outpatient clinical pharmacy is also explored in this field and garners interest from patients. The vast majority of studies demonstrate significant improvement in patient care with promising performance outcomes when pharmacists are involved. This review highlights the diverse range of interventions by clinical pharmacists in rheumatology, which is very promising. However, to better assess the benefits of clinical pharmacists, this activity needs further development and evaluation through controlled and randomized clinical research programs.

## 1. Introduction

Rheumatology is a complex discipline that encompasses several pathologies known for their significant morbidities. Among these, rheumatoid arthritis (RA) is the most common autoimmune pathology, affecting 1% of the global population. Additionally, various types of treatments are used in this discipline, requiring prescribers to master treatments with narrow therapeutic margins, such as colchicine or methotrexate, as well as innovative therapies with high clinical and health economic stakes involving the use of biologics and their biosimilars.

The updated definition from the European Society of Clinical Pharmacy (ESCP) specifies that «clinical pharmacy encompasses both a scientific discipline and a professional practice […], clinical pharmacy primarily aims to optimise the achievement of health-related goals in ways that support human dignity and personal choice (“person-centered goals”). The focus is on optimising the utilisation of medicines and implies that its goals are to be achieved via maximising medication effectiveness and safety. Person-centered goals aim to acknowledge patient preferences even when they might be at odds with evidence based practice. Public health goals, i.e., those that benefit society as a whole, can for example be achieved through prudent use of antibiotics to avoid microbial resistance. The revised definition avoids explicit mention of economic outcomes, which are considered of secondary importance to health-related goals. Although not explicitly mentioned by the revised definition (to reduce complexity), it is acknowledged that the achievement of person-centered goals may be constrained by public health goals and available resources» [1].

This discipline is increasingly gaining interest in various specialties, namely in diabetes, oncology [2], surgery [3], geriatrics [4], renal transplantation [5], and orthopedics [6]. The clinical pharmacist helps optimize patient care by improving medication adherence, a key factor for the success of treating chronic diseases [7,8]. The work by Gil-Guillen et al. notably demonstrates that information and education are essential to enhancing treatment adherence in all patients [8].

According to the team of Rowley et al. [9], pharmacists can intervene at different levels within the medical team responsible for managing a rheumatology patient, both in community and hospital settings. The national survey by McLachlan et al., conducted among adult patients with osteoarthritis, reveals inadequate relief in 73% of patients, while less than 20% of participants reported receiving recommendations from a community pharmacist. These findings advocate for the need for health information and education, and consequently, for the development of ambulatory clinical pharmacy activities to be conducted at the counter by pharmacists [10]. The team of Sah et al. [11] specifically investigated the impact of the clinical pharmacist on the clinical and health-economic aspects of managing patients with autoimmune diseases, including those with RA. The results of this study were positive, with 80% of the studies showing improvement in at least one clinical parameter due to pharmacist care and 80% demonstrating significant improvement in treatment adherence in the group receiving pharmacist-led care during the follow-up period. Clinical pharmacy plays an essential role in the specialty of rheumatology, optimizing medication management and improving patient adherence. This role can transform treatment paradigms by integrating personalized pharmaceutical interventions that specifically target the needs of patients with rheumatic diseases.

The main objective of this work is to identify the activities and involvement of the clinical pharmacist in rheumatology, as well as their clinical and economic impact on patient care. This research aims to assess the clinical and economic impacts of clinical pharmacists in rheumatology practice from 2000 to June 2024.

## 2. Materials and Methods

### 2.1. Search Strategy

The literature review conducted follows the PRISMA guidelines [12]. The bibliographic search for the study was carried out using the PubMed^®^, CINAHL^®^, Cochrane Library^®^, and Web of science^®^ databases and was last conducted on the 3 June 2024, using the following keywords: clinical pharmacy, pharmacist, pharmaceutical care, rheumatology, arthritis. For each of these databases, the search was executed using the following combination: “pharmacist AND (clinical pharmacy OR pharmaceutical care) AND (rheumatology OR arthritis)”. A restriction to the publication period was applied starting from the year 2000. Additionally, we conducted a search using the database of an unindexed French-language journal specialized in clinical pharmacy, namely the Journal de Pharmacie Clinique. Finally, the search was complemented by a manual search, utilizing the Impact Pharmacie^®^ database.

The search was independently conducted by two clinical pharmacists (CL and EB) with at least 1 year of experience in rheumatology service. Any discrepancies in selection were discussed, and inclusion or exclusion was decided upon consensually. Likewise, data extraction from the various studies was carried out by the two clinical pharmacists (CL and EB).

### 2.2. Study Selection

The inclusion criteria were as follows:▪Studies conducted specifically in the rheumatology sector or concerning a clearly identified rheumatic pathology.▪Studies describing pharmaceutical care activities, including the following [13]:-Medication reconciliation;-Medication review;-Pharmaceutical intervention;-Personalized pharmaceutical plan (PPP).▪Studies in both the French and English languages. ▪Abstracts for conferences, study protocols, and articles not describing specific clinical pharmacy activities (reviews, meta-analyses, etc.) were excluded.

### 2.3. Study Evaluation

The quality and risk of bias of the selected studies were assessed by the two clinical pharmacists (CL and EB) using the same procedures as the study selection (the search was independently conducted by two clinical pharmacists, any discrepancies in the selection were discussed, and inclusion or exclusion was decided upon consensually). We abided by the guidelines suggested by The EQUATOR (Enhancing the Quality and Transparency of Health Research) Network corresponding to the type of study:For randomized studies: the CONSORT (Consolidated Standards of Reporting Trials) checklist.For observational studies: the STROBE (Strengthening the Reporting of Observational studies in Epidemiology) checklist.For qualitative studies: the SRQR (Standards for Reporting Qualitative Research) checklist.For health economic studies: the CHEERS (Consolidated Health Economic Evaluation Reporting Standards) checklist.

The EQUATOR Network (Enhancing the Quality and Transparency of Health Research) is the first coordinated effort to assess the quality of scientific studies. This network provides reporting guidelines for the main study types. These guidelines, compiled by the EQUATOR Network, allow for the objective evaluation of all our studies using a standardized source. Thi is why quality assessments like CONSORT, STROBE, SRQR, and CHEERS from the EQUATOR Network were used.

Any discrepancies in the evaluations between the two pharmacists were discussed to reach a consensus. For each study, a proportion of fulfilled criteria was assigned. The results will be presented as general averages accompanied by their confidence intervals (M ± CI).

Quality analysis was conducted using the GRADE (Grading of Recommendations Assessment, Development, and Evaluation) method, which assigns four levels of quality to studies: high, moderate, low, and very low.

Randomized studies are of high quality but may be downgraded in cases of bias and depending on the study quality; similarly, observational studies are of low quality but may be downgraded in cases of quality issues or upgraded in cases of exceptional quality and/or highly significant results. Thus, in this work, we used the evaluations of each study according to the EQUATOR guidelines to downgrade or upgrade the studies using the GRADE method.

## 3. Results

The initial search identified 285 references. According to the various inclusion criteria, 24 articles were selected for this analysis (Figure 1).

These studies were conducted in Europe [14,15,16,17,18,19,20,21,22,23,24,25,26] (56.5%), with 26.1% of them specifically in France [14,15,19,20,22,23], 34.8% in North America [27,28,29,30,31,32,33,34], 26.1% in the United States [27,28,29,31,33,34], one involving international cooperation [35] (4.3%), and another in Jordan [36] (4.3%).

Among the selected articles, 12 were observational studies [14,16,20,22,23,26,28,30,31,32,34,36], 2 were mixed-method protocol studies [25,26], 8 were randomized studies [17,18,19,21,26,29,33,35], and 1 was a health economic study [15]. There is a trend towards an increase in research, with 78% of the studies published after 2010.

The studies in this review cover a diverse geographical range, particularly in Europe and North America, reflecting different approaches to pharmaceutical care in rheumatology. European studies highlight the effectiveness of pharmacists’ interventions in medication reconciliation and economic impact, while North American studies emphasize their role in managing drug-related problems and improving treatment adherence. Compared to experimental studies, cohort and case–control studies show significant long-term clinical improvements, whereas randomized controlled trials demonstrate statistically significant improvements in treatment adherence and patient satisfaction.

### 3.1. Medication Reconciliation and Medication Review

Various studies focus on pharmaceutical analysis and medication reconciliation in rheumatology [14,15,22,23], which are presented in Table 1. All those found during our literature search were conducted in France. However, these studies were of rather good quality, with a proportion of 72% (±12%) of the EQUATOR scale criteria being met.

Regarding the acceptance of pharmaceutical interventions (PIs) by prescribers, it is high for each study, ranging from 67.2% [23] to 93.3% [22].

For three of these studies, a prescription review was conducted. The rating scales used were those of the French National Authority for Health (HAS) [37]. The results were similar, as rheumatologists identified 43.8% of at least significant pharmaceutical interventions (PIs) in the Soubieux et al. team [22] compared to 56.2% in the Yailian et al. team [23]. Furthermore, 41.4% of unintentional discrepancies identified during medication reconciliation were judged to be at least significant in the Soubieux et al. team [22] compared to 49.6% according to a physician–pharmacist pair in the Boursier et al. team [14]. The main pharmacist interventions in these activities involved dosage adjustments, changes in administration modalities, or omissions of treatment upon hospital admission.

In addition, the articles of Soubieux et al. [22] and Boursier et al. [14] highlight that polymedication is a relevant criterion for prioritizing patients for clinical pharmacy activities.

Finally, the team of Coursier et al. examined the economic benefits of this activity [15]. They showed that the pharmaceutical analysis of half of the prescriptions in a rheumatology service allows for a saving of EUR 5940 for the institution.

### 3.2. Hospital Pharmaceutical Interventions

Several studies focus on pharmaceutical consultations conducted in hospitals [19,20,27,30,34,36,37]. Furthermore, among the topics discussed during the consultation, various common themes emerge: medication adherence, patients’ knowledge, the proper use of medication, side effects of treatments, etc. Moreover, the work conducted by Gutermann et al. shows a significant improvement in patients’ knowledge after six months of biotherapy treatment for those who benefited from the pharmaceutical consultation (*p* < 0.001) [19]. It is also observed that pain in rheumatology and its consequences, such as the misuse of opioids, NSAIDs, or other analgesics, is a theme that has been gaining more attention in recent years, as evidenced by the studies of Lattard et al. [20] and Issa et al. [36]. The publications found on this subject are described in Table 2.

### 3.3. Ambulatory Pharmaceutical Interventions

The majority of the articles described pharmaceutical interventions in community pharmacies [16,17,18,21,24,25,26,28,29,31,32,33,35]. The conditions discussed included rheumatoid arthritis in four studies, gout in three studies, arthritis or osteoarthritis in three studies, rheumatological or musculoskeletal disorders in two studies, and low back pain. The overall results are presented in Table 3. It can be noted that medication adherence is a concept explored in 64% of these studies [21,24,25,26,29,31,33]. Other themes include therapeutic education and the identification of pharmacotherapeutic problems. The study by Marra et al. [32] stands out as it highlights the role of community pharmacists in identifying and screening patients suffering from arthritis.

### 3.4. Studies’ Quality

Overall, ¾ of the quality criteria corresponding to the study type were met (77% ± 5%). To assess the quality of randomized studies, we used the CONSORT scale. There were eight randomized studies. These studies were of good quality (81% ± 7%). However, there was wide variability in the quality of the studies, ranging from 41% to 93%, even though five out of the eight studies met more than 85% of the CONSORT criteria. The details of the study quality are presented criterion by criterion in Table 4. According to the GRADE method, six of these studies were of high quality, one of moderate quality, and one of low quality.

For observational studies, we used the STROBE criteria. For the 14 observational studies, the quality ranged from 55% to 84%, with an average of 76% ± 5%. A representation of the criteria that were met or not met is presented in Table 4.

There was only one health economic study for which we used the CHEERS criteria. This study met 61% of the applicable criteria and was classified as a study of very low quality according to the GRADE methodology. A summary of the classification of studies is presented in Table 4.

## 4. Discussion

This work reflects the diversity of clinical pharmacy practices in the field of rheumatology. It is interesting to note that in all studies, pharmaceutical actions complement rather than oppose other medical or paramedical actions. Thus, studies with comparators compare a pathway with a pharmacist to a pathway without. This demonstrates the profession’s spirit of adapting to the needs of patients and healthcare teams, always seeking to integrate into care pathways. The activities are therefore very diverse, but can also be integrated into numerous care pathways with different actors and scopes that can adapt to local needs.

First of all, the quality of the studies is rather good. Our evaluation by study category shows that nearly three-quarters of the quality criteria are met for the selected studies. It is also interesting to note that pharmacists participate in the management of various rheumatological conditions (rheumatoid arthritis, gout, musculoskeletal disorders, osteoarthritis, arthritis, pain), with interventions ranging from medication reconciliation [14,15,22,23] to more targeted activities such as patient education actions [16,18,19,20,21,24,25,26,28,29,30,31,32,33,34,36], personalized pharmaceutical plans [20], diagnostic assistance [32], or therapeutic monitoring [29,31,33,37]. These interventions can be carried out in outpatient or hospital settings.

Medication reconciliation at admission or discharge has generally demonstrated its benefits, enabling safe and optimized patient management, as well as reducing morbidity and costs to society [34,36,38]. Medication reconciliation, described in numerous international publications, proves useful in various specialties, such as emergency medicine [39] or geriatrics [40,41], to reduce medication errors. In the field of rheumatology, several publications highlight the advantages of medication reconciliation. Indeed, in a moderate-quality study, the team of Soubieux et al. indicates the presence of numerous unintentional discrepancies identified during admission or discharge medication reconciliation. The authors indicate that 30.9% of unintentional discrepancies are significant and 7.6% are major.

In addition to reconciliation, it is observed that pharmaceutical interventions are very often used to address various rheumatological issues in outpatient or hospital settings. Nineteen studies (69.5%) concern this activity [16,17,18,19,20,21,24,25,26,27,28,29,30,31,32,33,34,35,36], of which 14 were published after 2015. Moreover, the results of studies on this activity seem reliable, as all studies classified as high quality concern pharmaceutical interventions. Pharmaceutical interventions have the advantage of being able to adapt to specific issues in very diverse pathologies. They seem to be clinically relevant and adapted to various rheumatological conditions, with a positive impact of this type of pharmaceutical care observed in the different articles. The effectiveness criteria range from satisfaction to improving quality of life. Indeed, the team of Grech et al. showed a significant improvement in the quality of life of patients (*p* < 0.05) in the context of RA [18]. An improvement in patient knowledge was observed in two studies [19,20]. Thus, in rheumatology, conducting pharmaceutical interventions is an interesting avenue for improving patient management. The clinical pharmacist plays an important role in securing and streamlining the patient care pathway.

In the 2000s, the advent of biomedicines considerably improved the management of patients with RA. However, for the optimal efficacy and safety of these treatments, it is essential to educate patients to optimize medication adherence. According to the literature, there is indeed a lack of adherence in RA [42]. Factors favoring non-adherence are notably related to the patient’s understanding and knowledge of the disease and treatment, their beliefs, and the time available with healthcare professionals [43,44,45]. In this review, medication adherence was evaluated in eight studies [21,24,25,27,29,31,33,34] (30.4%), of which five were comparative studies [21,25,29,31,33]. It is observed that the clinical pharmacist tends to improve patient medication adherence, with two studies showing a significant improvement in this parameter [21,33] and three others noting a non-significant but rather favorable trend towards the pharmacist [24,29,31]. Improving treatment adherence and reducing iatrogenesis through pharmacists’ interventions are crucial for better clinical outcomes in rheumatology. However, obstacles persist, such as variability in health policies, the need for interprofessional collaboration, and resistance to change. Overcoming these challenges requires policy reforms, enhanced professional training, and the development of collaborative practice models to fully integrate clinical pharmacy services into rheumatology care.

Most treatments prescribed in rheumatology are available on an outpatient basis. Thus, community pharmacists also play a crucial role in the healthcare journey of patients with rheumatological conditions. This work suggests that pharmacist-managed educational programs in an outpatient setting have positive effects for the patient. Various limiting factors are mentioned by community pharmacists, including the need for training, organizational issues, or a lack of connection with the hospital. However, they recognize their potential role and the expected benefits of this therapeutic education [46,47]. These benefits were also observed in a study conducted in Bulgaria, which analyzed the effects of an educational program led by community pharmacists for patients with gout. This work reported better pain management (significant decrease in crisis frequency after intervention in the intervention group; *p* = 0.001), a significant decrease in adverse effects in the intervention group (*p* < 0.001), and a significant decrease in visits to the general practitioner in the intervention group (*p* = 0.003), as well as the frequency of emergency medical calls (*p* = 0.001) [21]. Similar results were observed in two other Californian studies [29,33]. Additionally, a Canadian study reported another possible role for community pharmacists in screening for certain conditions. Thus, using a questionnaire, the pharmacist could identify over 80% of patients with previously undiagnosed knee osteoarthritis, allowing for the early management of these patients [32].

From an economic point of view, some works have shown interesting effects of pharmacists, in that they limit the number of patients not taking their medications due to economic problems related to insurance [27] or through direct savings [15]. However, this work highlights the lack of studies evaluating the medico-economic impact of clinical pharmacy in this discipline. For example, in the work of Lattard et al. (low quality), a trend towards a decrease in morphine treatment time was observed in patients who received care from clinical pharmacists [20]. Such an approach could be associated with a decrease in healthcare costs, but it has not yet been evaluated.

Many works focus on clinical pharmacy practices in rheumatology. However, our study reveals a need for progress in clinical research in this field. Indeed, among the 24 articles analyzed, only 10 (35%) used a comparator [18,19,20,21,24,29,31,33,37], of which 6 (26%) were randomized [18,19,21,24,29,33] and only 4 were of high quality [19,21,24,33] and 1 of moderate quality [29]. This methodological weakness constitutes bias for these studies, and the clinical and economic impact of the clinical pharmacist still needs to be demonstrated in rheumatology through additional randomized studies.

Although informative, this review presents certain limitations. First, there is publication bias; despite conducting the broadest search possible, the published studies may not accurately represent all research conducted, as studies with negative results are less likely to be published. Additionally, there is heterogeneity among the designs. Different inclusion criteria and varied methodologies make the direct comparison of results between studies difficult. The studies included in this systematic review reveal several inadequacies that require a critical evaluation of the validity of the results. A frequently observed limitation is the diversity of methodologies used, particularly the often small sample sizes, that could limit the generalization of the conclusions [14,16]. Furthermore, significant variations in inclusion criteria and evaluation methods have been identified, which can introduce heterogeneity into the collected data [22,23]. Some reports have also raised concerns about potential biases, such as selection or publication bias, which may influence the reported results [24,27]. Finally, the variability in measures used to evaluate clinical outcomes sometimes makes it difficult to directly compare the effects of the studied interventions [29,30]. These inadequacies underscore the importance of a rigorous and transparent methodological approach in interpreting the conclusions of this review, as well as the need for future research to address these limitations and strengthen the evidence base in the field of clinical pharmacy in rheumatology.

## 5. Conclusions

The main findings of this review demonstrate that integrating clinical pharmacy services into rheumatology care significantly enhances the quality of care and clinical outcomes for patients. Pharmacist interventions enable the detection and correction of medication errors and the identification and resolution of drug-related problems, and contribute to the better management of musculoskeletal disorders. To effectively integrate clinical pharmacy services into rheumatology care pathways, it is recommended to formalize the role of pharmacists within multidisciplinary teams, including their active participation in medication management, patient education, and care coordination. Developing continuous training programs for pharmacists is also crucial to ensure they have the necessary skills to intervene effectively. Policy changes are needed to support these recommendations. It is essential to establish health policies that recognize and value the role of clinical pharmacists, as well as ensure adequate funding for these services. Additionally, standardizing clinical pharmacy practices through national guidelines is important to ensure uniform quality of care. These measures will help implement the recommendations and improve clinical outcomes for patients with rheumatic diseases. Furthermore, it is important to emphasize that these efforts must be conducted within a multidisciplinary practice. The integration of clinical pharmacy services should complement the expertise of other healthcare professionals, fostering a collaborative environment that maximizes patient care. Multidisciplinary collaboration is crucial in ensuring comprehensive and coordinated care for patients, leading to better health outcomes and more efficient use of healthcare resources.

## Figures and Tables

**Figure 1 healthcare-12-01463-f001:**
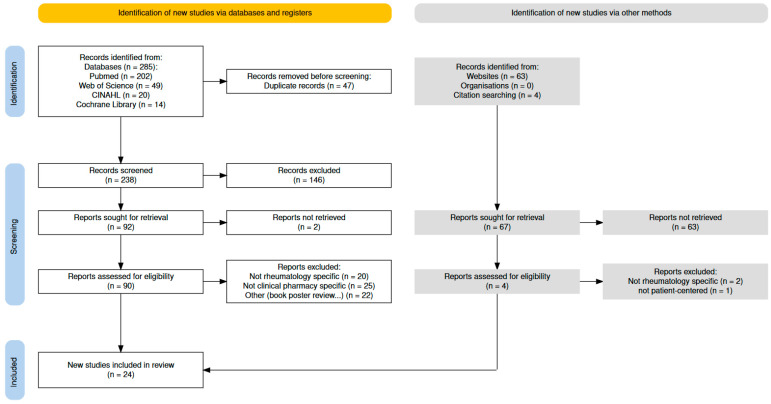
Flow chart.

**Table 1 healthcare-12-01463-t001:** Reconciliation pharmaceutical analysis and medication review.

References	Number of Patients	Design	Main Objective	Number of MDs * (AR **%)	PI *** (AR **%)	Field	Impact (HAS Scale)	Comments
Boursier et al. [14]	318	Observational study	Assessing the value of medication reconciliation in admission to a university rheumatology department	248 (100%) 1.2 ± 1.62 DVNI/ patient	NK	Omission (54%) Dosage (39.9%)	At least 51.6% MDs meaningful 2% major	Impact of pairing of rheumatologist and pharmacist Prioritization criterion Potential polypharmacy
Soubieux et al. [22]	334	Observational study	Impact of the presence of a pharmacist on drug iatrogenicity	At admission: 385 (93%) At discharge: 132 (95%) 1.66 DVNI/Patient	196 (93.3%)	PI: dosage (38.8%) MD: omission (55% at admission and 46% at discharge) and dosage (27%)	* PI: 30.9% meaningful and 12.9% major * MD: 33.8% meaningful and 7.6% major	Impact of rheumatologist Prioritization criterion Potential polypharmacy All physicians and patients found pharmacist intervention useful
Yailian et al. [23]	373	Observational study	Evaluating the impact/relevance of pharmaceutical interventions from the point of view of a pharmacist and a rheumatologist	NK	461 (67.2%)	Mode of administration (31.7%) Dosage (24.5%)	* Physician: 10% major and 1PI critical * Pharmacist: 20% major and no PIs critical	Impact of pairing of rheumatologist and pharmacist
Coursier et al. [15]	Not specified	Cost minimization analysis	Cost savings through pharmacist intervention	NA	610 (83.4%)	Mode of administration (25.4%)	NA	Cost saving: EUR 5940

* MD: medication discrepancies; ** AR: acceptance rate; *** PI: pharmaceutical intervention.

**Table 2 healthcare-12-01463-t002:** Pharmaceutical interventions in hospital.

References	Number of Patients/Methodology/ Main Objective	Theme/Disease/ Pharmaceutical Intervention	Main Results	Comments
Peter et al. [34]	776/ Retrospective observational study/ Assess adherence, persistence, and number of switches	SC * biotherapy/ Rheumatoid arthritis/ Identify and resolve obstacles to adherence	Adhesion and persistence superior to the given literature	Observational retrospective study
Hall JJ et al. [30]	62/ Observational study/ Patient satisfaction	SC * biotherapy/ Inflammatory arthritis/ Medication therapy management	Improved general patient satisfaction and giving information in the group with collaborative pharmacist–physician management (significant) No significant difference in the mean score for satisfaction across the 6 dimensions	The pharmacist evaluates the efficacy and tolerance of the biotherapy + evaluation of the disease with clinical and biological scores
Petry et al. [27]	480/ Retrospective cohort study/ Reduction in primary non-compliance	SC * biotherapy/ Inflammatory rheumatism/ Medication therapy management and resolving treatment-related issues such as insurance coverage	Primary non-adherence is much lower in this study with the clinical pharmacy service than in the literature (primary non-adherence: 2.1%)	This decrease is attributed to the dedicated pharmacy service, which considers not only clinical aspects (patient counseling) but also financial aspects (ensuring the patient is insured).
Romano et al. [37]	124/ Retrospective cohort study/ Achieve and maintain serum uric acid of less than 6 mg/dL	Uricemia follow-up/ Gout/ Follow-up of patients on hypouricemia therapy	Higher proportion of patients reaching the uricemia target in the pharmacist group (75.8% vs. 30.6%) than in the physician group. For patients reaching the target, the speed was identical in the 2 groups, but with a lower dose for the pharmacist group.	NA
Issa et al. [36]	100/ Cross-sectional before–after study/ Pain relief	Transdermal glucosamine sulfate and capsaicin/ Joint pain/ Medication therapy management	Significant decrease in pain, limitation of mobility, use of other analgesics, and number of visits to the doctor Increase in mild and moderate stiffness and decrease in severe to moderate stiffness	NA
Lattard et al. [20]	35/ Non-randomized interventional study/ Improved patient knowledge	Opioid drug/ Pain in rheumatology/ Medication therapy management	Decrease in incorrect beliefs and increase in knowledge about opioid drugs Non-significant decrease in duration of opioid treatment (33.5 for control group vs. 14.0 for intervention group)	Pharmaceutical intervention at initiation of strong opioid therapy Result not significant, probably due to lack of patients
Guttermann et al. [19]	89/ Randomized single-center study/ Improved knowledge and better MPR scores	SC * Biotherapy/ Ankylosing spondylitis/ Medication therapy management	Improved knowledge at 6 months (significant vs. control) Tendency (non-significant) to improve the already high level of membership No impact in disease activity All patients satisfied	A study conducted over a short period and with few patients

* SC: subcutaneous.

**Table 3 healthcare-12-01463-t003:** Ambulatory pharmaceutical interventions.

References	Number of Patient/Methodology/Main Objective/Pharmaceutical Intervention	Main Results
Ernst et al. [16]	461/ Observational study/ Identify drug-related problems using a health status questionnaire/Medication review	good capacity of the pharmacist to identify and act on drug-related problems good correlation between drug-related problems and quality of life impairment
Ernst et al. [28]	461/ Observational study/ Assess the link between drug-related problems and the quality of life of patients suffering from musculoskeletal disorders/ Quality of life questionnaire, health status monitoring, and medication-related issues	926 drug-related problems identified in 461 patients. 2 types of problems with a significant association with negative change in MCS: poor treatment and untreated indication (item also significantly correlated with MCS)
Grech et al. [18]	88/ Observational study/ Assess personalized pharmaceutical care in a specialized rheumatology clinic/ Medication therapy management	Setting up of pharmaceutical consultations to identify problems in the pharmaceutical sector in patients treated with methotrexate, with delivery of a booklet on this treatment. 84% of patients found the methotrexate booklet useful, some even said that such an initiative would be useful for other drugs. 106 pharmaceutical problems were identified in 88 patients: 72% were real problems that required a change in treatment. Concerning quality of life: significant improvement in SF36 and HAQ scores after intervention
Marra et al. [32]	194/ Observational study/ Diagnosis of osteoarthritis by community pharmacists/ Screening for arthritis using a questionnaire	Pharmacists in pharmacies can identify osteoarthritis in patients with chronic knee pain: 83% of patients diagnosed with osteoarthritis by pharmacists using a questionnaire with arthritis 82% of these patients without prior diagnosis Faster support
Huang et al. [31]	47/ Comparative study without randomization/ Pharmaceutical intervention to improve adherence and effectiveness of gout treatment/ Pharmaceutical care	Results similar to those of Goldfien: 31% of the intervention group reached the target zone for uricemia at one year (*p* = 0.3%)
Mikuls et al. [33]	1463/ Randomized study/ Pharmaceutical intervention to improve adherence and effectiveness of gout treatment/ Phone call	Patient intervention group: better adherence than the control group (50 vs. 37% *p* < 0.001) More patients in the intervention group reached the target uricemia rate (30 vs. 15% *p* < 0.001)
Petkvova et al. [21]	86/ Randomized study/ Quality of life of patients/ Outpatient therapeutic education	After education: significant improvement for the intervention group in pain frequency vs. control group; significant improvement in medication adherence; significant decrease in emergency calls in the intervention group vs. before education (no difference for the control group); significant decrease in visits to the attending physician and number of adverse events after education in the intervention group (not in the control group).
Zwikker et al. [24]	123/ Randomized study/ Improvement of adherence of RA patients/ Motivational interviews	No difference between the two groups. Only after one year did patients in the intervention group have fewer incorrect beliefs about their treatment than in the control group.
Goldfien et al. [29]	77/ Randomized study/ Improve adherence and effectiveness of gout treatment/ Pharmaceutical care (prescription of laboratory tests or therapeutic adjustment by pharmacist)	In total, 35% of patients in the intervention group (13/37) reached the target level of 6 mg/dL at 26 weeks, compared to 13% in the control group (but no significant difference) Greater decrease in uricemia in the intervention group than in the control group (singular)
Thapa et al. [35]	158/ Randomized study/ Change in pain levels and improvement in physical functionality/ Education and medication review	The pharmaceutical intervention improved patients’ knowledge and pain scores; however, physical functionality, depression, and quality of life remained unchanged
Hay et al. [17]	325/ Randomized study/ Change in pain levels/ Review of medication and patient information	The pharmacist helped reduce pain similarly to the physiotherapist, and there was slightly lessened functionality, but better than the control group, at 3 months. In the long term, there was no significant difference except for the physiotherapist, who reduced visits to the primary care physician.
Zwikker et al. [25]	228/ Qualitative study/ Motivational interviews to improve adherence of RA patients	Review of the literature: the problem of adherence Cross-sectional study: association between incorrect beliefs and non-adherence Discussion group: non-adherent patients have many incorrect beliefs Implementation of a motivational interview with the pharmacist. Pilot: patients appreciated the intervention (even if they found it a bit short)
Wilbur et al. [26]	5/ Mixed method: group focus and semi-structured individual interviews/ Presence of a clinical pharmacist on the team	According to all members of the team making up the rheumatology specialty clinic, the pharmacist’s presence is useful for education regarding the treatments prescribed in rheumatology: role in improving patient understanding and adherence. Medication reconciliation desired by the team members Full-time pharmacist desired

**Table 4 healthcare-12-01463-t004:** Studies’ quality.

References	Study Type	EQUATOR Network’s Guideline Used	Proportion of Positive Criteria	Quality Level (GRADE)
Marra et al. [32]	Observational	STROBE	82%	Moderate
Hall et al. [30]	Observational	STROBE	79%	Low
Zwikker et al. [25]	Observational	STROBE	68%	Low
Soubieux et al. [22]	Observational	STROBE	83%	Moderate
Yailian et al. [23]	Observational	STROBE	83%	Moderate
Ernst et al. [16]	Observational	STROBE	79%	Low
Huang et al. [31]	Observational	STROBE	68%	Low
Boursier et al. [14]	Observational	STROBE	63%	Very low
Peter et al. [34]	Observational	STROBE	73%	Low
Issa et al. [36]	Observational	STROBE	83%	Moderate
Ernst et al. [28]	Observational	STROBE	55%	Very low
Lattard et al. [20]	Observational	STROBE	77%	Low
Petry et al. [27]	Observational	STROBE	84%	Moderate
Romano et al. [37]	Observational	STROBE	86%	Moderate
Mikuls et al. [33]	Randomized	CONSORT	93%	High
Goldfien et al. [29]	Randomized	CONSORT	74%	Moderate
Petkova et al. [21]	Randomized	CONSORT	77%	High
Grech et al. [18]	Randomized	CONSORT	41%	Low
Zwikker et al. [24]	Randomized	CONSORT	92%	High
Gutermann et al. [19]	Randomized	CONSORT	88%	High
Thapa et al. [35]	Randomized	CONSORT	93%	High
Hay et al. [17]	Randomized	CONSORT	88%	High
Wilbur et al. [26]	Qualitative	SRQR	81%	Moderate
Coursier et al. [15]	Cost effectiveness	CHEERS	61%	Very low

## Data Availability

Not applicable.

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
