# Peer review of "Impact of the Clinical Pharmacist in Rheumatology Practice: A Systematic Review"

_healthcare, 2024, doi:10.3390/healthcare12151463_

Round 1

Reviewer 1 Report

Comments and Suggestions for Authors

This a well written paper that meets an important need in pharmacy. I think it deserves publication but there are two important issues that need to be addressed before publication.

#1. Literature Review.  The literature review for this paper went through the end of 2021 so it is now 2½ years old. Given that the literature on this topic is rapidly expanding, as shown in figure 2, that means the authors have probably missed at least 8 to 10 recent papers of relevance! It would take considerable work but if the literature review were updated it would be a much more useful paper.

On a related note, did the authors have access to International Pharmaceutical Abstracts? Since this database specializes in pharmacy literature, it may also add to the reference list.

#2.  Terminology. The article title says impact in the rheumatology ward. To my American ears, this implies interventions in a hospital ward, yet most of the studies are in ambulatory care. Thus, I strongly suggest changing the term to rheumatology PRACTICE.

Many of the studies are classified as “pharmaceutical interviews.” I find this a strange classification that implies simple face-to-face discussion with a patient or provider. In fact, most of these are studies of pharmacy INTERVENTIONS and I strongly suggest using that terminology.

More broadly, the classification of papers seems odd to me, as each classification and table contains quite disparate articles. For example, pharmaceutical analysis implies something like a chemical analysis to me. In this group, one study on cost-minimization is thrown in with value of medication reconciliation. I’m not sure of the best way to group the studies, but separating randomized intervention studies from observational studies should be considered.

In the tables there is no consistent description of the intervention. For example, in table 2 the first study examines impact of patient education and the second study looks at collaborative management as the intervention, yet that cannot be known without reading the citations. Thus, the second column of each table needs a better description of intervention.

Minor issues

The abstract says nothing about improved outcomes for patients. I believe this the most important endpoint.

Lines 51-52 in the introduction, don’t forget the very large literature on pharmacy interventions in diabetes and hypertension.

In table 2 the abbreviation SC is not defined

Comments on the Quality of English Language

Minor issues discussed in author comments.

Author Response

Dear reviewer,

Thank you for your careful reading and your comments. Please find enclosed our responses.

Kind regards.

The authors.

Reviewer 2 Report

Comments and Suggestions for Authors

Thank you for your submission. I believe major changes are needed before the manuscript could be considered for international publication.

The scientific writing and English needs improvement e.g. even the title is grammatically incorrect.  Phrases with “etc.” should not be used in scientific writing e.g. “…gaining interest in various specialties, namely in oncology [2], surgery [3], geriatrics [4], renal transplantation [5], orthopaedics [6] etc…”.  That statement is also incorrect. The role of clinical pharmacy in fields such as oncology and geriatrics has been well established for many years. Similarly, sentences ending in “……” (line 275) are not used in research manuscripts.

The discussion is far too long. It should be no more than half its current length. Scientific writing should be concise, especially when the study had relatively few studies included and they were generally low quality – each study does not warrant a verbose description.

This sentence makes no sense: “From an economic point of view, although the pharmacist's intervention during pharmaceutical consultations focused on biologics could have a major impact, although not yet objectively demonstrated, a study looked at the economic impact of pharmaceutical interventions conducted in a rheumatology and functional rehabilitation service.”

The use of a single database (PubMed) needs justification. Cochrane is not applicable and serves a different purpose, while the non-indexed French database does not count.  Other relevant databases should have been used (e.g. IPA, Embase, CINAHL).

Why does the title refer to “ward”?  The review and search strategy seems to cover both hospital and community settings.

Figure 2 should be deleted. It is unnecessary and the key information is stated in the text. Similarly, Figures 3 to 5 should be deleted. The key information is already in the text and tabulated. The figures are also confusing.

What is meany by “pharmaceutical analysis”? In English, this term refers to analysing the content of pharmaceutical preparations or dosage forms in a laboratory. It is not a clinical activity.

“Drug iatrogenicity” is another term not used in English.

Why are studies that “..focused specifically on the expectations of community pharmacists to develop their clinical pharmacy skills” included? That is irrelevant to the topic and research question.

Comments on the Quality of English Language

Extensive revision of the English is required, commencing with the title - (i) grammatically incorrect and also (ii) factualy incorrect

Author Response

(The authors gave the same response as above.)

Reviewer 3 Report

Comments and Suggestions for Authors

General Comments To address how clinical pharmacy influences patients, this systematic review identifies the general contribution that clinical pharmacy makes within the center. However, several areas of the manuscript need improvement to increase clarity, rigor, and overall impact.

Title and Abstract Title:

-          The title is informative and does not waste words. Consider defining the review's scope in more specific terms—for instance, geographically, if appropriate.

  • Suggestion: "Impact of Clinical Pharmacists in the Rheumatology Wards across Europe and North America: A European and North American Study Review"

Abstract:

-          The abstract overall does an excellent job of making the review purpose of this study precise but should ideally include a few more minor methodologies and critical findings.

  • Suggestion: Here, the number of studies reviewed, the significant metrics analyzed such as medication adherence, and cost-savings and precise recommendations for future research should be given.

Introduction

  • Remark: It is a good introduction since it provides an insight into the background of the research, thereby setting the stage for clinical pharmacy significance in rheumatology. Just that the research can be more effective by presenting a clear research problem, the research questions, and then objectives.
  • Comment: Good, since it has provided a background to the significance of clinical pharmacy in rheumatology—very clear research question at the very end of the introductory section. For example, "the research aims to assess the clinical and economic impact of clinical pharmacists in the rheumatology wards from 2000 to 2021."

Methods

-Search strategy well stated; to make this more rigorous, grey literature sources could be included.

  • Suggestions: Indicate if grey literature or non-indexed journals were included. Include a statement about what was done about restrictions on language how studies over different periods were included, and why that time frame was chosen.

Study Selection and Evaluation:

  • Comment: The eligibility criteria for a study to be included and quality of studies is adequate, but there is a lack of explicit justification
  • Suggestion: More fully explain why you decided to use these quality assessment tools about CONSORT, STROBE, SRQR, and CHEERS.

Results

  • Comments on Results: Results have been very elaborate but written clumsily to some extent. The recommendation would be that the help of sub-headings can manage the result.
  • Recommendation: Orient the section in terms of sub-headings, for example, "Pharmaceutical Analysis and Medication Reconciliation," "Pharmaceutical Interviews," and "Economic Impact." Use summary tables for each category to complement the readability.

Figures and Tables

  • Comment: Figures 1 and 2 are okay and informative but they could contain better descriptions and legends.
  • Recommendation: All figures and tables should have an appropriate title, and caption must be at least descriptive. A summarized table of the main results of each study should be incorporated to make the review informative.

Discussion

- Of course, the discussion is well done concerning incrementally reflecting on the implications of the findings. However, increasing the critical appraisal concerning the limitations and the varying qualities of the studies might make it more coherent.

-           It would be recommended to add a section addressing the limits of the review, such as publication bias, heterogeneity among the designs, and inadequacy of the underlying studies. Trying to address specific areas for future research is hence suggested, and comparison with some more RCTs would be desirable.

Conclusion

-          Finally, the conclusion is relatively short, and some additional sentences could be adjoined to, for example, indicate some of the key findings from the review as well as its clinical and policy implications.

-          Implication: Key study findings to be highlighted and implications of these for clinical practice; clear recommendations as to how clinical pharmacy services can be integrated within the rheumatology care pathway; what policy changes are needed to realize the recommendations

References

-          The references should be: supplied in total, but it is essential to make sure that references to those that are quoted in the text have been cross-checked—suggestion: Reviewed to ensure its currency and relevance. This paper could be enriched by including more recent reviews or meta-analyses that would provide some context to this work.

Specific Comments on Sections

  • Introduction: Elaborate more on how clinical pharmacy plays a role in the rheumatology specialty and how that role can be used to change treatment paradigms.
  • Methods: Describe data extraction methodology clearly and how differences of opinion between reviewers have been resolved
  • Results: Elaborate on the distribution of studies geographically and what it could mean. Compare and contrast significant differences in outcomes for the cohort and case-control studies compared with the outcomes for the experimental studies.
  • Discussion: Elaborate on the clinical importance of increased medication adherence and reduced iatrogenesis. Predict the barriers to full clinical pharmacy practice for each of the following:
  • Conclusion: Describe interdisciplinary patient management approaches in rheumatology conditions. Describe the implementation of a policy directive for the widespread availability of clinical pharmacists as part of the multidisciplinary approach to rheumatology care.
Comments on the Quality of English Language

Moderate editing of English language required

Author Response

(The authors gave the same response as above.)

Round 2

Reviewer 1 Report

Comments and Suggestions for Authors I congratulate the authors on a very thorough update to their paper. I have just one major issue left.   I was surprised the updated literature review did not catch more articles, so I checked my own files and found 2 more recent ones. Their PMID numbers are 35012933 and 36549929. Perhaps they were found but excluded for various reasons so I will leave it to the authors whether to include them in the final version.    Minor issues:  On the 6th page, lines 186 to 188. The paragraph beginning with "Moreover, some of these studies..." does not make sense to me. It sounds like the methods were based on the results.   Similarly, the third sentence in the discussion "The only study comparing showed comparable results" is nonsensical.   In the discussion, line 312, the authors refer to 24 studies instead of 23.   Table 3 has some typos for "Observational". Comments on the Quality of English Language

Minor typos.

Author Response

Thank you for your attentive and valuable review. Here are our responses to your latest comments:

Comment 1: I was surprised the updated literature review did not catch more articles, so I checked my own files and found 2 more recent ones. Their PMID numbers are 35012933 and 36549929.

Response 1: We confirm that these two articles were not found in our search equation. Both are very interesting. After a thorough review of these two articles, PMID 35012933 was not included because the study did not evaluate clinical impact on patients. However, study 36549929 is highly relevant to our research. Thank you very much for your assistance. This study has been included, and the inherent modifications have been made (flow chart, bibliography order, etc.).

Comment 2: Minor issues: On the 6th page, lines 186 to 188. The paragraph beginning with "Moreover, some of these studies..." does not make sense to me. It sounds like the methods were based on the results. Similarly, the third sentence in the discussion "The only study comparing showed comparable results" is nonsensical. In the discussion, line 312, the authors refer to 24 studies instead of 23. Table 3 has some typos for "Observational".

Response 2: We have corrected all the minor issues as requested. Once again, thank you for your help.

Sincerely,
The Authors

Reviewer 2 Report

Comments and Suggestions for Authors

Thank you for the revision. It has improved substantially.

Comments on the Quality of English Language

Some of the use of English terms is odd, but understandable.

Author Response

dear reviewer,

Thank you for your attentive reading

The authors

Reviewer 3 Report

Comments and Suggestions for Authors

Many thanks for your efforts and commitment to reply all comments.

Comments on the Quality of English Language

Minor edits can be corrected in the proofreading stage.

Author Response

(The authors gave the same response as above.)

Round 3

Reviewer 1 Report

Comments and Suggestions for Authors

All concerns have been addressed.